# *Limosilactobacillus fermentum* MG4244 Protects Against Metabolic and Inflammatory Stress in *Caenorhabditis elegans*

**DOI:** 10.3390/foods14111995

**Published:** 2025-06-05

**Authors:** Yebin Kim, Opeyemi O. Deji-Oloruntoba, Yunji Choe, Jiyeon Lee, Jeongyong Park, Byoungkook Kim, Sooim Choi, Miran Jang

**Affiliations:** 1Department of Institute of Digital Anti-Aging Healthcare, Inje University, Gimhae 50834, Republic of Korea; dpqls1812@gmail.com; 2Biohealth Convergence Unit, Food and Drug Biotechnology, Inje University, Gimhae 50834, Republic of Korea; oodejioloruntoba@gmail.com; 3Department of Food Technology and Nutrition, Inje University, Gimhae 50834, Republic of Korea; 29280@naver.com; 4MEDIOGEN, Co., Ltd., Biovalley 1-ro, Jecheon 27159, Republic of Korea; ljy341@naver.com (J.L.); pjy@mediogen.co.kr (J.P.); kbk@mediogen.co.kr (B.K.); csi@mediogen.co.kr (S.C.)

**Keywords:** MG4244, intestinal permeability, reactive oxygen species, lipid accumulation, AMPK, MAPK

## Abstract

In this study, we investigated the effects of MG4244 on intestinal permeability, oxidative stress, and lipid accumulation in *Caenorhabditis elegans* with metabolic inflammation induced by *Pseudomonas aeruginosa* (PA) and a high-glucose diet (HGD). The worms infected with PA exhibited increased intestinal permeability and reactive oxygen species (ROS) production, which were improved upon MG4244 treatment. Also, MG4244 inhibited lipid and ROS accumulation induced by an HGD. In addition, MG4244-treated worms showed extended lifespans under various conditions. To elucidate the mechanism of the MG4244 effects, we conducted further investigation using mutant strains with knockdown of genes associated with the AMP-activated protein kinase (AMPK) and mitogen-activated protein kinase (MAPK) pathways. The results demonstrated that the MG4244 effect on lipid metabolism was primarily mediated through the AMPK signaling pathway. Furthermore, MG4244 enhanced pathogen resistance by MAPK signaling pathways, mitigating stress responses, and maintaining intestinal integrity. In further studies, combined treatment with PA and an HGD significantly increased intestinal permeability, lipid, and ROS levels, confirming their negative synergistic effects. However, MG4244 under PA and HGD co-treatment conditions effectively mitigated these health disruptions, suggesting a protective role of MG4244. This study provides an in vivo platform using *C. elegans* to evaluate probiotic efficacy related to the intestinal environment. Also, our results highlight the therapeutic potential of MG4244 in improving resilience to metabolic inflammation through gut-targeted mechanisms.

## 1. Introduction

The intestinal barrier is a complex system of microbiota, immune cells, a mucus layer, and epithelial cells, connected by tight junctions [1]. When these tight junctions are disrupted, intestinal permeability increases, resulting in leaky gut syndrome, a condition associated with poor nutrient metabolism, infections, stress, and chronic inflammation [2].

Probiotics provide health benefits to the host by modulating the gut microbiota, strengthening the intestinal barrier, enhancing immunomodulation, and regulating metabolism in vivo [3]. These properties make probiotics effective in managing conditions such as irritable bowel syndrome (IBS) and inflammatory bowel disease (IBD) by alleviating symptoms like diarrhea and bloating, restoring the balance of the gut microbiota, reducing inflammation, and promoting mucosal healing [4]. In particular, their capacity to improve glycemic control, regulate fat metabolism, enhance insulin sensitivity, and modulate appetite has made probiotics valuable in managing metabolic disorders and obesity [5]. Probiotics have demonstrated the potential to mitigate leaky gut by targeting key metabolic pathways, such as AMP-activated protein kinase (AMPK) and mitogen-activated protein kinase (MAPK) [6,7]. AMPK activation enhances the expression of tight junction proteins, thereby improving the integrity of the epithelial barrier [8]. Additionally, probiotics can modulate the mTOR pathway, which regulates lipid synthesis and insulin signaling, as well as influence MAPK pathways to reduce inflammation and oxidative stress [9].

Among probiotics, *Limosilactobacillus fermentum* is a gram-positive facultative anaerobe commonly found in the human gastrointestinal tract, vaginal microbiota, and fermented foods such as yogurt, sourdough, and pickles [10]. Several strains of *L. fermentum* have been shown to enhance the intestinal barrier by responding to oxidative stress and inflammation [11,12,13]. Additionally, these strains are well recognized for their beneficial effects on energy metabolism, which occur through various molecular pathways [14,15,16]. Meanwhile, different strains of *L. fermentum* exhibit strain-specific activities, underscoring the importance of selecting the appropriate strains for targeted therapeutic applications [17]. The MG4244 strain has demonstrated stronger adhesion to intestinal epithelial cells (Caco-2 and HT-29) compared with the CECT5716 strain, which exhibits only moderate adhesion. MG4244 also shows notable antioxidant and anti-inflammatory properties, although such data remain limited for other strains. Additionally, it possesses strong bacteriocin-like inhibitory activity against pathogenic organisms. MG4244 has been shown in previous studies to regulate lipid metabolism through AMPK both in vitro and in vivo, thereby inhibiting obesity [18,19,20,21]. However, the therapeutic benefits of MG4244 in improving intestinal permeability have not yet been reported.

This study aimed to investigate the relationship between intestinal permeability and metabolic disorders using *Caenorhabditis elegans*, with a particular focus on the effects of MG4244. Worms were pretreated with MG4244 for a specific period, followed by exposure to *Pseudomonas aeruginosa* or glucose (GLU) to induce intestinal permeability or metabolic disorders. This approach allowed us to assess the response of the worms to these conditions. The primary objective was to evaluate the protective effects of MG4244 in a model of an intestinal permeability-induced metabolic disorder and to elucidate the underlying molecular mechanisms.

## 2. Materials and Methods

### 2.1. Sample Preparation

The *Limosilactobacillus fermentum* MG4244 used in this study was provided by MEDIOGEN (MEDIOGEN Co., Ltd., Jecheon, Republic of Korea). MG4244 was cultured in MRS broth at 37 °C for 24 h. The *Pseudomonas aeruginosa* (PA) used in this study was provided by American Type Culture Collection (ATCC10145; Manassas, VA, USA). PA was cultured in Luria–Bertani broth (LB broth, by Difco, Sparks, MD, USA) at 37 °C using an incubator. Before the experiment, the bacterial culture was adjusted to an optical density of 0.2 at 600 nm (OD600) using M9 buffer.

All reagents used in our study were HPLC or molecular biology grade. Unless stated otherwise, all the materials were purchased from Sigma Chemical Co. (St. Louis, MO, USA).

### 2.2. Worm Cultivation

We used the *C. elegans* N_2_ strain (wild-type) and its derivative mutant strains; daf-16 (tm5030), atgl-1 (tm12352), aak-1 (tm1944), akt-1 (tm399), pmk-1 (tm13288), sek-1 (tm14334), and skn-1 (tm4241), all of which were obtained from the National BioResource Project (NBRP) of Tokyo, Japan.

All strains were maintained on nematode growth medium (NGM) plates seeded with *Escherichia coli* OP50 and kept at 20 °C throughout the experiment. Age synchronization of the nematodes was achieved by isolating the eggs from gravid adults using a solution containing 6% sodium hypochlorite and 5 M NaOH [21].

### 2.3. Acute Toxicity

Synchronized L4 larvae were washed twice with M9 buffer and subsequently maintained in M9 buffer supplemented with 5% cholesterol. One milliliter of the suspension was transferred to each well of a 24-well plate (containing 25–30 worms per well) and mixed with 10 μL of MG4244 at varying concentrations. The plates were incubated at 20 °C for 24 h. Acute toxicity was assessed by calculating the survival rate as a percentage based on the number of live worms.

### 2.4. Measurement of Intestinal Permeability

Intestinal permeability analysis was conducted with modifications to the method outlined by Kim and Moon [22]. In summary, age-synchronized young nematodes cultured on OP50 were grown to young adults (YAs) and subsequently exposed to either OP50 or MG4244 for 1 day. Subsequently, the nematodes were exposed to either OP50 or PA for 1 day before being used in the experiments. The adult nematodes were then treated with FD&C Blue (5% *w*/*v*) for 3 h. The nematodes were washed five times with buffer and examined under a microscope to evaluate whether the dye had diffused beyond the intestine.

### 2.5. Oil Red O Staining

To evaluate lipid accumulation in *C. elegans*, the Oil Red O (ORO) assay was used [21]. The L1-stage worms were cultured until they reached the YA stage and subsequently exposed to either OP50 or MG4244 each day. Afterward, the worms were exposed to either OP50 or 2% GLU for the experiment.

Cultivated worms were fixed in 4% formaldehyde for 24 h and then dehydrated with 60% isopropanol at −70 °C for 15 min. The dehydrated worms were washed three times with M9 buffer and stained with an ORO solution for 2 h. The stained worms were washed again with M9 buffer and observed under a Nikon ECLIPS Ci microscope (Nikon, Seoul, Republic of Korea). The relative intensity of the stained lipid droplets in the worms was quantified using ImageJ software (version 1.8.0). Approximately 15 worms per group were selected for quantification.

### 2.6. Measurement of Reactive Oxygen Species (ROS) Levels

YA worms were exposed to either OP50 or MG4244 for one day, followed by treatment with OP50 or *P. aeruginosa* for an additional day. The treated worms were then incubated in the dark with 100 μM H_2_DCF-DA for 3 h. After that, the nematodes were mounted in NAN_3_ (2%) onto microscope slides. The slides were viewed using a Nikon ECLIPS Ci microscope, Tokyo, Japan. The fluorescence intensities were examined using Image J software. Approximately 15 worms per group were selected for quantification.

### 2.7. Determination of Stress Resistance

To assess stress tolerance, fifty synchronized YA worms were prepared by feeding either OP50 or MG4244 for one day. For the heat stress assay, worms were initially maintained at 20 °C for 60 h before being exposed to 35 °C for 32 h.

For the oxidative stress assay, prepared worms were transferred to 24-well plates containing a 100 μM juglone (5-hydroxy-1,4-naphthoquinone) solution and incubated at 20 °C. We then analyzed the mean lifespan of each worm population (fifty worms per group) subjected to these stress conditions to evaluate their survival and stress tolerance.

### 2.8. Lifespan Assay

Various concentrations of extracts were prepared, including 120 µM 5-Fluoro-2′-deoxyuridine (FUDR, 98%; Alfa Aesar, Seoul, Republic of Korea) to inhibit reproduction and 50 µg/mL carbenicillin to prevent bacterial contamination. The transfer day was designated as day 0, during which the old medium was replaced with a fresh medium containing the extracts. This procedure was repeated every other day until all the worms had perished. Survival rates were expressed as percentages (% survival rate).

### 2.9. Statistical Analyses

Statistical analyses were conducted on data obtained from multiple replicates. Differences between groups were assessed using one-way analysis of variance (ANOVA) followed by Tukey’s multiple range test for post hoc comparisons. All statistical analyses, excluding lifespan-related data, were performed using SPSS version 27.0. Lifespan, mean lifespan, and stress resistance data were analyzed using the Kaplan–Meier method with the OASIS application (https://sbi.postech.ac.kr/oasis/, accessed on 21 July 2024). Survival differences were evaluated using the log-rank test, with a *p*-value of <0.05 indicating significance.

## 3. Results

### 3.1. Safety of MG4244

Acute toxicity tests were performed to evaluate the safety of MG4244. The experimental results indicated that MG4244 exhibited toxicity at a concentration of 1 × 10^9^ CFU/mL, while its safety was confirmed at a concentration of 1 × 10^7^ CFU/mL (Figure 1). Considering this result, all subsequent experiments were conducted at a concentration of 1 × 10^7^ CFU/mL.

### 3.2. MG4244 Modulates PA- and HGD-Induced Health Deterioration in C. elegans

To evaluate the impact of MG4244 on intestinal function, we measured intestinal permeability using established protocols. In this study, the ND group was fed the standard OP50 diet, another control group was exposed to PA, and the sample-treated group received MG4244 pretreatment before PA exposure. Healthy worms showed dye confined to the intestine and intestinal lumen, whereas worms with compromised gut integrity displayed dye diffusion into the body cavity or other tissues. Our results indicated that in the ND group, body fluid was distributed between the intestine and the intestinal lumen in a 40:60 ratio, with no fluid observed elsewhere in the body. In contrast, in the PA-treated group, approximately 60% of the body fluid shifted from the intestinal area into the entire body, suggesting leakage into the body cavity. However, upon MG4244 ingestion by wild-type worms, we observed an improvement in body-cavity leakage (Figure 2A). We further evaluated the therapeutic effect of MG4244 pretreatment on ROS levels in worms infected with PA. The results showed that the PA group exhibited high levels of ROS accumulation, which were significantly reduced in the MG4244-treated group (Figure 2B).

To evaluate the therapeutic effect of MG4244 on HGD-induced metabolic abnormalities, *C. elegans* were treated with 2% glucose as well as with MG4244 in order to examine its impact on lipid accumulation. The results showed that MG4244 reduced lipid accumulation in worms to levels similar to those observed in the ND group (Figure 2C). Also, our results showed that MG4244 reduced ROS levels to levels comparable to those of the ND group under HGD conditions (Figure 2D).

Taken together, these findings suggest that MG4244 ameliorates PA- and HGD-induced physiological disruptions.

### 3.3. Effect on the Lifespan of MG4244 Under Normal Conditions

Lifespan experiments conducted with the control group and the MG4244-treated group revealed no significant difference. However, the mean lifespan of the control group was 14.92 ± 0.36 days, while that of the MG4244-treated group was 15.56 ± 0.52 days, indicating a tendency for an increase in the MG4244-treated group (Figure 3).

### 3.4. Effect on the Lifespan of MG4244 Under Oxidative and Thermal Conditions

In this study, N_2_ worms were divided into a control and an MG4244-treated group. Following oxidative stress induction with the oxidative inducer juglone, the MG4244-treated group exhibited a significant increase in oxidative stress resistance (Table 1). The mean lifespan of the control group was 5.39 ± 0.56 days, while the MG4244-treated group had a mean lifespan of 13.81 ± 1.21 days, representing an approximately 256% increase in lifespan. However, no significant difference in mean lifespan was observed between the two groups under thermal stress conditions.

### 3.5. The Effect of MG4244 on Improving Lipid Metabolism Is Mediated by AMPK Related Factors

To identify the genetic factors involved in the lipid metabolism-improving effect of MG4244, the lipid levels of the daf-16, aak-1, akt-1, and atgl-1 mutants were also assessed. The results confirmed that the inhibitory effect on lipid accumulation of MG4244 observed in the N_2_ worms was reduced or abolished in the atgl-1, aak-1, daf-16, and akt-1 mutant strains (Figure 4).

### 3.6. PA-Induced Intestinal Permeability Improvement Effect of MG4244 Is Related to Sek-1 and Daf-16

The intestinal permeability assay was conducted under identical conditions in the N_2_ worm strain and all the mutant strains as used in this study. We observed that body-cavity leakage was reduced when the N_2_ worms were fed MG4244. To further investigate the metabolic pathways involved in body-cavity leakage, we the employed aak-1, daf-16, akt-1, atgl-1, pmk-1, sek-1, and skn-1 mutant worms to assess the roles of these pathways. Among the tested strains, the intestinal permeability-enhancing effect of MG4244 observed in the N_2_ worms was largely diminished in the daf-16 and sek-1 mutants (Figure 5G). However, in the remaining strains, the overall trend remained consistent with that observed in the N_2_ worms (Figure 5). Specifically, in the ND group, fluid was primarily confined to the intestine and intestinal lumen. In contrast, in the PA-treated group, fluid spread extensively throughout the body. In the MG4244-treated group, fluid remained localized within the intestine and surrounding intestinal regions.

### 3.7. The Effect of MG4244 on Improving Oxidative Stress Is Mediated by AMPK, MAPK, and Related Factors

To investigate the genetic factors involved in the antioxidant effect of MG4244, ROS levels were assessed in mutants knocked down for key factors of the AMPK and MAPK pathways. The ROS reduction observed in the N_2_ strain was abolished in the mutants lacking AMPK-related genes (aak-1, daf-16, and atgl-1), except for akt-1 (Figure 6A–D). Further analysis identified skn-1, sek-1, and pmk-1 as critical components of stress response pathways, including the MAPK pathway and oxidative stress regulation. Notably, in these mutants, the ROS reduction effect was either significantly diminished or absent, compared with the effect observed in the N_2_ strain (Figure 6E–G).

### 3.8. MG4244 Improves Intestinal Permeability, ROS, and Lipid Levels Under Combined Treatment with PA and an HGD

PA increased intestinal permeability and ROS production but did not significantly induce lipid accumulation. In contrast, the high-glucose diet (HGD) promoted lipid accumulation and ROS production but had no significant effect on intestinal permeability. Interestingly, the combined treatment with PA and the HGD resulted in significantly elevated levels of intestinal permeability, ROS production, and lipid accumulation compared with treatment with PA or the HGD alone. However, despite the significant increase in symptoms indicative of metabolic disease with the co-treatment of PA and the HGD, MG4244 was found to significantly improve intestinal permeability, ROS levels, and lipid accumulation under these conditions (Figure 7).

## 4. Discussion

Our previous study demonstrated that MG4244 improves lipid metabolism in 3T3-L1 cells and obese mice [18,19]. However, this study aimed to explore the previously unreported effect of MG4244 on intestinal permeability.

Our results indicated that MG4244 positively affects intestinal permeability and alleviates PA-induced oxidative toxicity. Additionally, it exhibits beneficial effects on lipid accumulation and oxidative stress as induced by an HGD. Together, PA infection and an HGD have been shown to induce mitochondrial dysfunction and elevate ROS levels, contributing to cellular stress [23,24]. These findings highlight the shared mechanisms through which pathogens and metabolic disturbances exacerbate cellular damage. We also found that MG4244 increased worm lifespan by more than 200% under juglone-induced oxidative stress conditions. Our results emphasize the therapeutic effect of MG4244 in combating these common mechanisms, offering potential for treating oxidative stress and metabolic disorders.

In a previous study, MG4244 inhibited lipid accumulation by regulating adipogenesis through AMPK phosphorylation [18,19]. AMPK, a critical cellular energy sensor, regulates metabolism and energy homeostasis in many organisms, including *C. elegans* [25,26]. This study also confirmed that MG4244 improves lipid metabolism in an aak-1/AMPK-dependent manner. Additionally, factors related to the AMPK signaling pathway, such as akt-1/Akt, daf-16/FOXO, and atgl-1/ATGL, were investigated and found to be involved in MG4244-mediated lipid metabolism. Notably, these factors are closely associated with the insulin signaling pathway, suggesting that they play a role in the process by which glucose is converted into fat and accumulates in *C. elegans* under an HGD. Together, AMPK and Akt work synergistically but competitively, coordinating cellular metabolism and maintaining energy homeostasis [27]. Both kinases phosphorylate FOXO, but in different contexts. AMPK activates FOXO under energy stress, promoting catabolic processes like autophagy and fatty acid oxidation [28]. In contrast, Akt inhibits FOXO, favoring anabolic processes such as cell growth and lipid synthesis [29]. Although it is clear that each of these factors plays a significant role in improving lipid metabolism through MG4244, further studies are needed to determine whether their mode of action is inhibitory or activating in relation to the efficacy of MG4244.

The positive effect of MG4244 on intestinal permeability observed in N_2_ worms was also observed in the aak-1, atgl-1, akt-1, skn-1, and pmk-1 mutants, but it was not observed in the daf-16 and sek-1 mutants. PA interferes with host immune responses by activating the daf-2 insulin-like signaling pathway, which inhibits the daf-16/FOXO transcription factor. This inhibition reduces the expression of immune defense genes [30]. The intestinal expression of daf-16 is critical for resisting PA, and its suppression weakens the host’s immune defense capabilities. Therefore, maintaining balanced daf-16 activity is essential for preserving intestinal integrity and robust immune responses [31]. Furthermore, the p38 MAPK pathway is critical in the defense of *C. elegans* against PA infection [32]. In particular, the MAPK pathway is also required for defense against opportunistic species, with mutants in this pathway being more susceptible to infection [33]. As part of the p38 MAPK signaling pathway, sek-1 plays a role in the innate immune response of *C. elegans* [34]. Indeed, our research results confirmed that the improvement effect of MG4244 on intestinal permeability was weakened in the mutants lacking sek-1 and daf-16. According to Zheng et al. [35], intestinal immune regulation during PA infection in *C. elegans* was dependent on sek-1 and daf-16, similar to our findings. However, their study suggested that the dependence on daf-16 was more dominant than that on sek-1, which contrasts somewhat with our results. Accordingly, specific probiotic strains can differentially affect intestinal immunity in *C. elegans* by preferentially activating daf-16 or sek-1 pathways [25]. Many strains with anti-*Pseudomonas* effects tend to rely more on daf-16, aligning with the dominance observed in immune regulation studies. Meanwhile, Zhou et al. [36] reported that genes involved in the MAPK signaling pathway such as sek-1 and pmk-1 and the daf-16/FOXO signaling pathway showed no activation in response to the protective effects of *Lacticaseibacillus zeae* against Enterotoxigenic *E. coli* (ETEC)-induced infection in *C. elegans*. The AMPK and MAPK pathways are involved in maintaining immunity against pathogen infections and resisting stress, but the roles of specific components may vary depending on the pathogen species and even the specific strain.

As mentioned earlier, the mechanisms underlying the lipid accumulation inhibition and improvement of intestinal permeability by MG4244 are linked to the activation of AMPK and MAPK pathways, emphasizing the potential roles of these pathways in the antioxidant effects of MG4244. In addition, our results showed that when MG4244 was administered to the akt-1 mutants, a similar reduction in ROS levels was observed as in the N_2_ wild-type strain. The activity of akt-1 is known to promote ROS accumulation by inhibiting the antioxidant defense system. Conversely, low levels of akt-1 facilitate the translocation of daf-16 to the nucleus, which activates antioxidant gene expression, ultimately reducing ROS levels [37]. This phenomenon was also observed in our study.

We established four experimental models to investigate the mutual influence of inflammation and metabolic disorders: ND, PA-induced inflammation, HGD-induced metabolic disorders, and PA+HGD-induced inflammatory metabolic disorders. Our study confirmed the effects of MG4244 in this context. Interestingly, the combined treatment with PA and an HGD resulted in significantly higher lipid accumulation compared with an HGD alone. Moreover, the combination of PA and an HGD caused greater impairment of intestinal permeability and elevated ROS levels compared with either PA or an HGD treatment alone, indicating a negative synergistic effect. These results suggest that while PA alone does not directly induce obesity, it creates an inflammatory environment that, when combined with an improper diet such as an HGD, exacerbates lipid accumulation and oxidative stress. These findings emphasize the dynamic and interconnected processes involved in inflammation, oxidative stress, and metabolic disorders, emphasizing the need for targeted interventions in managing metabolic health. Furukawa et al. [38] suggest that increased oxidative stress in fat accumulation is an early instigator of metabolic syndrome and that the redox state in adipose tissue might be a promising therapeutic target for obesity-associated metabolic syndrome. Inflammation in the gut, along with disruptions to the gut microbiome and increased gut permeability (also known as “leaky gut”), is a condition primarily caused by factors such as bacterial infections. This allows endotoxins like lipopolysaccharides (LPS) to enter the bloodstream, triggering systemic inflammation and promoting fat accumulation [39]. In our study, we also confirmed that PA-induced inflammation induced ROS accumulation and increased intestinal permeability, which exacerbated obesity. However, these were alleviated by MG4244 administration.

Although inflammation and oxidative stress play key roles in obesity, the precise mechanisms by which chronic inflammation contributes to the development of obesity, as well as the pathways through which inflammatory cytokines are released in inflammatory obesity, remain poorly understood due to a lack of comprehensive biological data [40]. Therefore, further in-depth studies are needed to elucidate the developmental mechanisms underlying inflammation and metabolic diseases. Additionally, *C. elegans* is a useful model for studying oxidative stress and host–microbe interactions, but its simplicity limits translation to humans [41]. To bridge this gap, future validation should include the use of murine models to assess systemic effects and human intestinal cell lines (e.g., Caco-2, HT-29, T84, and HIEC-6) to study cellular responses, ensuring relevance to human health. Additionally, future studies should explore the efficacy of MG4244 on mammalian models and assess their interactions with the host microbiome, as this will add translational value. Another limitation of the study is that image analysis was not performed in a blinded fashion. Although this could introduce observer bias, we minimized variability by applying the same thresholding parameters, background correction, and region-of-interest selection criteria to all images.

## 5. Conclusions

In conclusion, MG4244 exhibited notable therapeutic efficacy in a model of inflammatory metabolic disorder induced by PA and a HGD. Our findings demonstrate that PA-induced intestinal inflammation significantly contributes to the progression of metabolic pathologies such as obesity, alongside intensifying oxidative stress and epithelial barrier dysfunction. MG4244 administration markedly improved intestinal barrier integrity, attenuated oxidative damage, and reduced lipid accumulation, primarily through modulation of the AMPK and MAPK signaling pathways. These mechanistic insights underscore the potential of MG4244 as a promising candidate for the prevention and management of inflammation-associated metabolic disorders. Nonetheless, the intricate molecular and genetic regulatory networks underpinning MG4244’s mode of action remain incompletely elucidated. Therefore, further in-depth studies are warranted to fully characterize its therapeutic potential and to explore its translational relevance in clinical settings.

## Figures and Tables

**Figure 1 foods-14-01995-f001:**
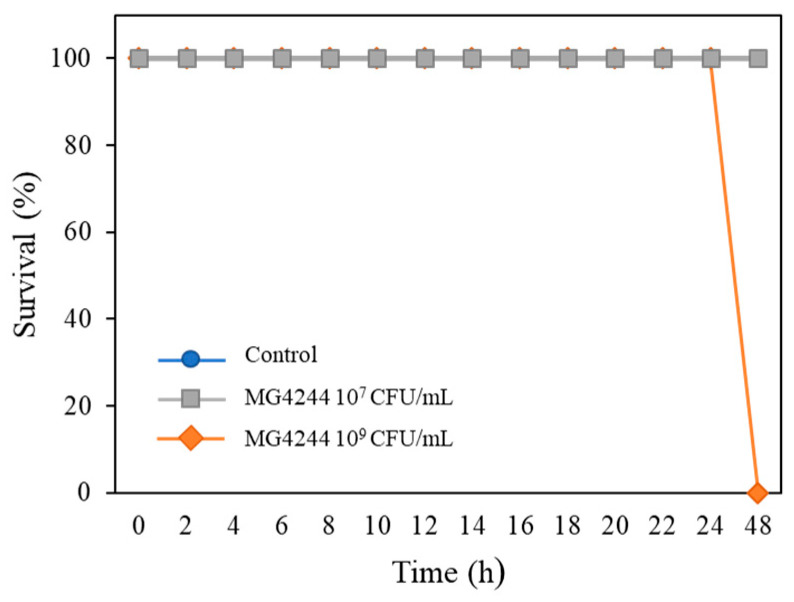
Survival of *C. elegans* under MG4244-treated conditions. The effect of MG4244 at different concentrations (10^7^ and 10^9^ CFU/mL). Approximately thirty worms were used in each group.

**Figure 2 foods-14-01995-f002:**
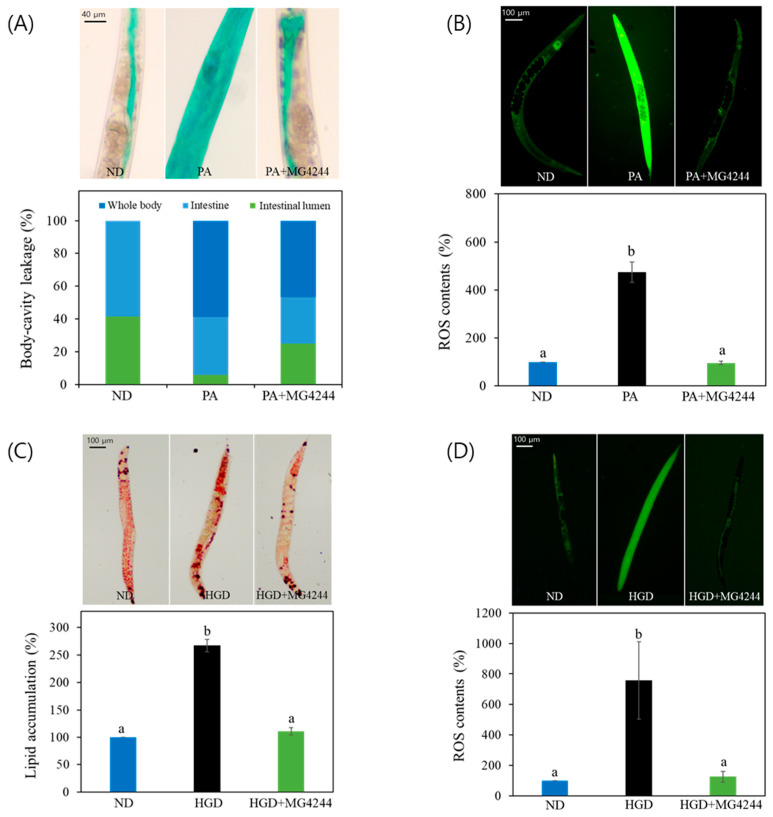
MG4244 reduces PA- or HGD-induced health deterioration in *C. elegans*. Effects on (**A**) intestinal permeability and (**B**) ROS increase in PA-treated worms and (**C**) lipid and (**D**) ROS increase in HGD-treated worms. Twenty worms were used in each experiment. Results are presented as mean ± standard deviation. Statistical analysis was carried out using one-way Anova. Different letters indicate statistically significant differences (*p* < 0.05, Tukey’s test). ND: Normal diet. PA: *Pseudomonas aeruginosa*. HGD: High-glucose diet.

**Figure 3 foods-14-01995-f003:**
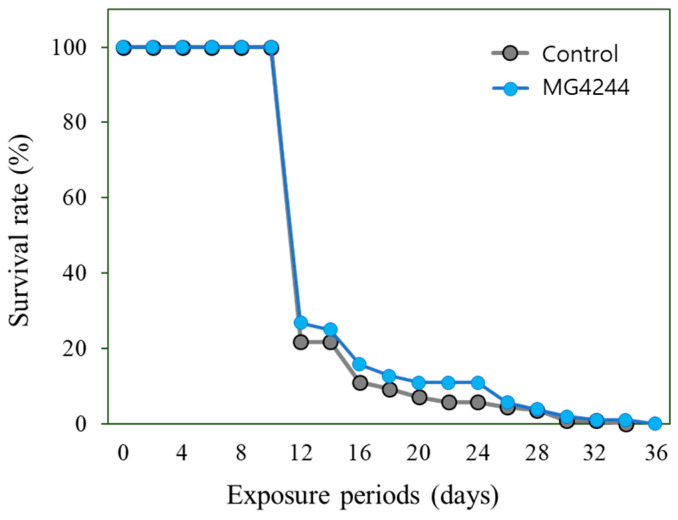
Effect of MG4244 on the lifespan of *C. elegans.* Approximately fifty worms were used in each group.

**Figure 4 foods-14-01995-f004:**
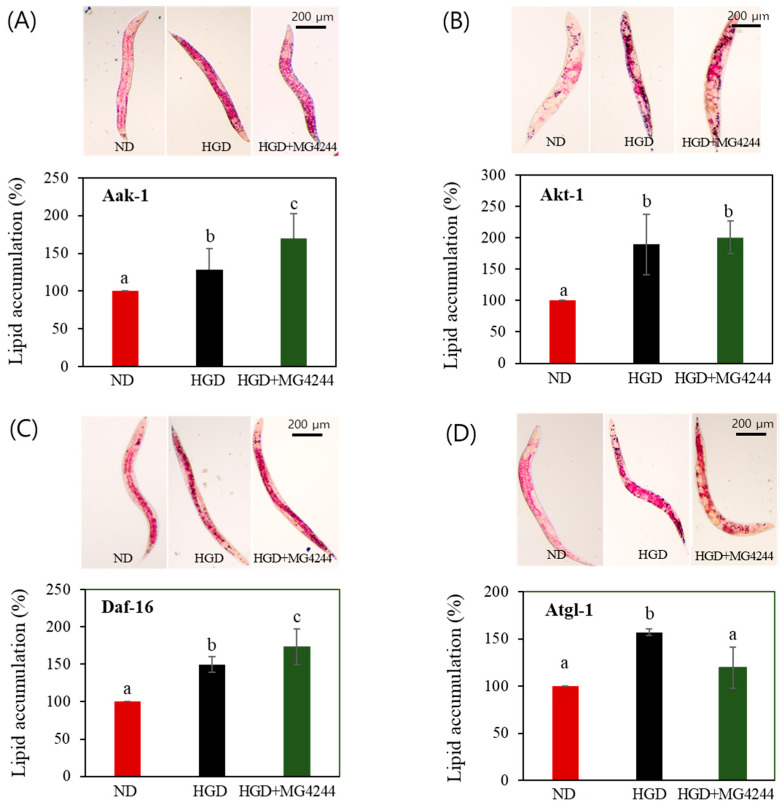
The effect of MG4244 on lipid accumulation depends on factors related to lipid metabolism. Images and quantitative bar graphs of total body lipids are shown for transgenic strains lacking four major lipid metabolism factors: (**A**) Aak-1, (**B**) Akt-1, (**C**) Daf-16, and (**D**) Atgl-1. Results are presented as mean ± standard deviation. Statistical differences among groups are indicated by different letters (*p* < 0.05). ND: Normal diet. HGD: High-glucose diet.

**Figure 5 foods-14-01995-f005:**
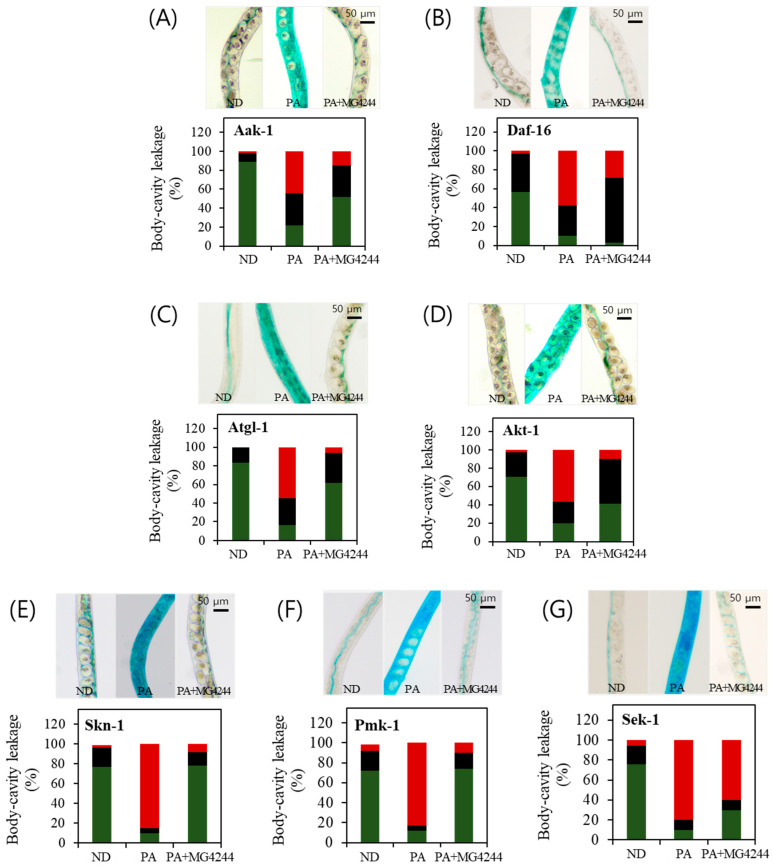
The effect of MG4244 on intestinal permeability is influenced by multiple genetic factors. PA was administered to induce intestinal permeability, and, in the MG4244-treated group, PA was applied after the completion of sample treatment. The images depict the distribution of blue pigments in the worms, specifically in the intestinal lumen, intestine, and whole body, with quantitative bar graphs representing these locations. (**A**) Aak-1, (**B**) Daf-16, (**C**) Atgl-1, (**D**) Akt-1, (**E**) Skn-1, (**F**) Pmk-1, and (**G**) Sek-1. ND: Normal diet. PA: *Pseudomonas aeruginosa*. The red, black and green colors represents: whole body, intestine and intestinal lumen respectively.

**Figure 6 foods-14-01995-f006:**
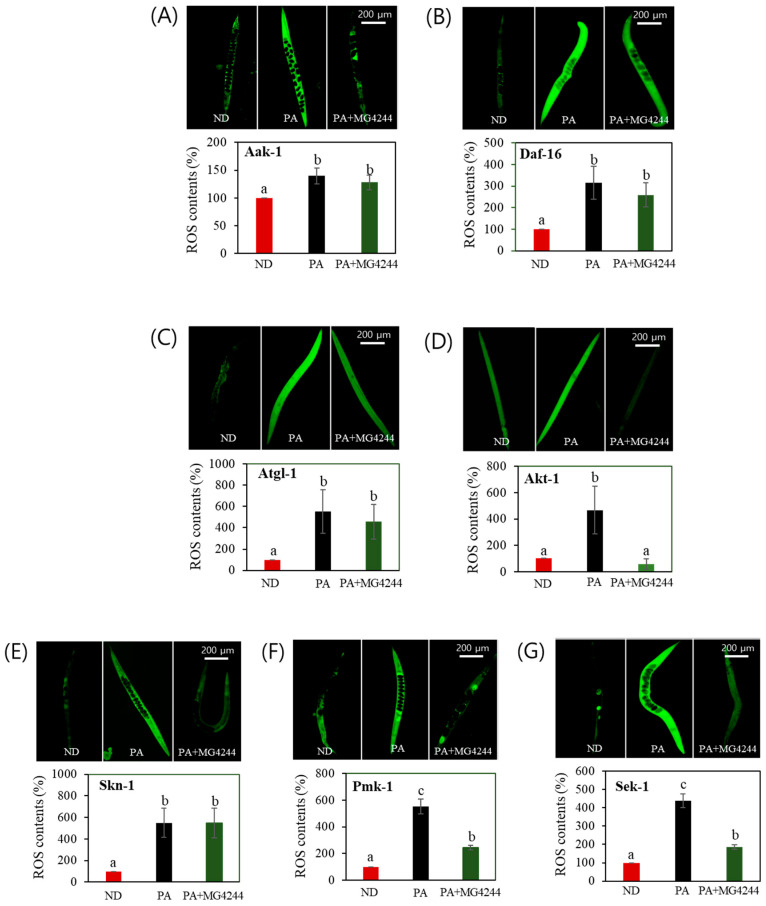
The effect of MG4244 on ROS accumulation depends on factors related to AMPK and MAPK. Images and quantitative bar graphs of total body ROS are shown for transgenic strains lacking four major lipid metabolism factors: (**A**) Aak-1, (**B**) Daf-16, (**C**) Atgl-1, (**D**) Akt-1, (**E**) Skn-1, (**F**) Pmk-1, and (**G**) Sek-1. Results are presented as mean ± standard deviation. Statistical differences among groups are indicated by different letters (*p* < 0.05). ND: Normal diet. HGD: High-glucose diet.

**Figure 7 foods-14-01995-f007:**
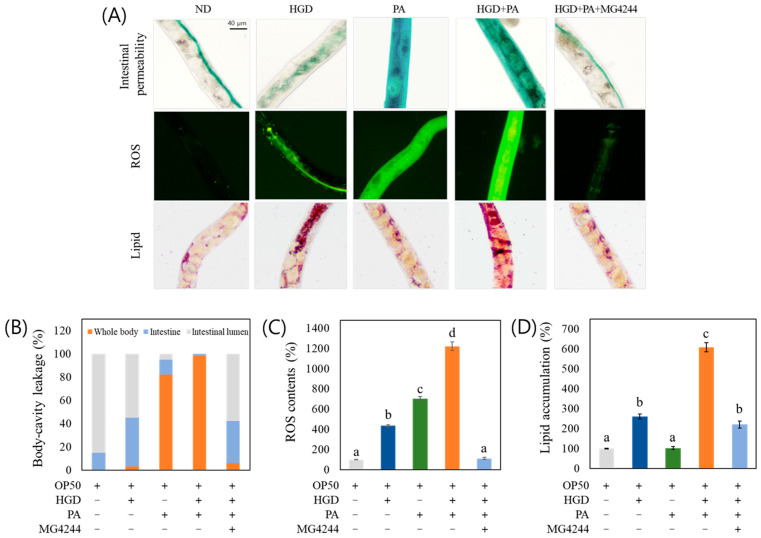
MG4244 reduces health deterioration in *C. elegans* under combined PA and HGD treatment. (**A**) Images of responses of worms to intestinal permeability, ROS levels, and lipid accumulation under various conditions. Quantitative graphs of (**B**) intestinal permeability, (**C**) ROS levels, and (**D**) lipid accumulation in worms under various conditions. Twenty worms were used in each experiment. Results are presented as mean ± standard deviation. Statistical differences among groups are indicated by different letters (*p* < 0.05). ND: Normal diet. PA: *Pseudomonas aeruginosa*. HGD: High-glucose diet.

**Table 1 foods-14-01995-t001:** Lifespan of MG4244-treated worms under oxidative and thermal stress conditions.

Conditions	Groups	Mean Lifespan (Days)	*p*-Value
Oxidative stress	Control	5.39 ± 0.56	<0.001
	MG4244	13.81 ± 1.24	
Thermal stress	Control	11.53 ± 1.28	0.8186
	MG4244	11.70 ± 1.09	

Results are presented as mean ± standard deviation. The OASIS application was used for the *p*-values presented here.

## Data Availability

The original contributions presented in this study are included in the article. Further inquiries can be directed to the corresponding author.

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
