# Peer review of "Limosilactobacillus fermentum MG4244 Protects Against Metabolic and Inflammatory Stress in Caenorhabditis elegans"

_foods, 2025, doi:10.3390/foods14111995_

Round 1

Reviewer 1 Report

Comments and Suggestions for Authors

This study was well designed and executed. However, there are a few minor issues that need to be addressed. 

Minor Comments

  • Line 80: Include the culture conditions for MG4244.

  • Line 85: "OD600" — make "600" a subscript (OD₆₀₀).

  • Line 95: Include the manufacturer or supplier information for NGM.

  • Line 104: Use either "hour" or "h" consistently throughout the manuscript; do not use both.

  • Line 108: Replace "In summary," with "Briefly,".

  • Statistical analysis: Revise to clearly indicate differences between ND vs. PA and PA vs. PA+MG4244 in Figure 2. Use asterisks to denote significance levels (e.g., p < 0.05, p < 0.01, p < 0.001). Apply this format consistently across all histograms for better clarity.

Author Response

REVIEWER 1

Comments and Suggestions for Authors

This study was well designed and executed. However, there are a few minor issues that need to be addressed.

Minor Comments

Comment 1

Line 80: Include the culture conditions for MG4244.

Response: MG4244 was cultured in MRS broth at 37°C for 24 h.

Line 85: "OD600" — make "600" a subscript (OD₆₀₀).

Response 2. The optical density has been written appropriately’

Line 95: Include the manufacturer or supplier information for NGM.

Response 3: Yes, we agree with this comment. However, there is a standard procedure for making NGM which involves the use of several other chemicals in the laboratory, we believe that adding such long details might not be necessary in the body of the work.

Line 104: Use either "hour" or "h" consistently throughout the manuscript; do not use both.

Response 4:  We have read through the manuscript and chosen one of the options “h”

Line 108: Replace "In summary," with "Briefly,".

Response 5: Yes, we agree with this comment, and we have changed it to “Briefly”

Statistical analysis: Revise to clearly indicate differences between ND vs. PA and PA vs. PA+MG4244 in Figure 2. Use asterisks to denote significance levels (e.g., p < 0.05, p < 0.01, p < 0.001). Apply this format consistently across all histograms for better clarity.

Response 5: We agree with this comment, but we believe that we stated it in line 170-173. “In this study, the ND group was fed the standard OP50 diet, another control group was exposed to PA, and the sample-treated group received MG4244 pretreatment before PA exposure”.

Secondly, we only tested at p<0.05, all through the study and this applies to all histograms

Reviewer 2 Report

Comments and Suggestions for Authors

The manuscript entitled “Limosilactobacillus fermentum MG4244 Attenuates Metabolic Inflammation-Induced Intestinal Permeability, ROS Overexpression, and Lipid Accumulation in Caenorhabditis elegans” presents solid experimental work and relevant findings. The use of mutant C. elegans strains effectively elucidates the roles of AMPK and MAPK pathways in MG4244’s protective effects.

The results support MG4244’s potential to reduce oxidative stress, lipid accumulation, and intestinal permeability under metabolic and inflammatory stress. While the study is generally well-designed and suitable for publication, several methodological and interpretive issues require clarification. Specific comments and questions are provided below to improve clarity and scientific rigor.

Major comments:

  1. lines 70–77: Did the authors assess whether MG4244 colonizes the gut of C. elegans, or are its effects due to transient exposure?If not, could this impact the interpretation of the long-term protective effects observed?
  2. lines 90–94 and 106–114: Was OP50 used as the control in all treatment comparisons, including PA and HGD groups?Clarifying this will help determine if dietary differences influenced outcomes independently of MG4244.
  3. lines 124–125 and 132–133: How were the thresholds and background levels determined in Image J for lipid and ROS analysis?Were the images analyzed in a blinded fashion to avoid bias?
  4. lines 150–153: Were the results based on biological or technical replicates?
    How many independent experiments were performed per treatment?
  5. lines 159–164: Was 1×10⁷ CFU/mL selected based only on toxicity, or was it confirmed to be the most effective dose?Was a dose–response study performed?

Minor comments:

1 lines 150–156: Could the authors report confidence intervals or effect sizes along with p-values to better represent effect magnitude?

  1. Would the authors consider simplifying the title for clarity and impact, such as “MG4244 Protects Against Metabolic and Inflammatory Stress in C. elegans”?
  2. lines 343–347: Can the authors elaborate on why their findings differ from Zheng et al. regarding the roles of daf-16 and sek-1*? Could strain-specific effects or differences in experimental setup explain this discrepancy?
  3. lines 391–400: Could the authors strengthen the conclusion by specifying what future studies should investigate (e.g., mammalian models, microbiome interactions)?
  4. The images are clear and appropriately labeled. However, resolution is low in some images (e.g., Figure 2A and 5 panels). Please ensure that high-resolution versions are submitted for publication. Consider increasing contrast and scale bar visibility to improve interpretability in microscopy panels.
  5. Several references (lines 410–512) have duplicated DOI links and visible formatting issues (e.g., line breaks, misplaced punctuation). These should be cleaned for consistency. Ensure all references follow the journal's required citation style (e.g., spacing, author names, italics for journal names).
  6. Some figure legends (e.g., Figures 2, 5, and 6) lack detail. Add the statistical test used, number of replicates (e.g., "n = 20 worms"), and what each letter or asterisk means for significance. For example: “Different letters indicate statistically significant differences (p < 0.05, Tukey’s test).”
  7. Line 407: The statement “available on request” is increasingly discouraged by many journals. Consider uploading raw data (e.g., fluorescence intensity values, lifespan curves) to a public repository (e.g., Zenodo, Figshare) and linking it.
  8. In bar graphs (Figures 2, 4, 6, 7), consider adding: Error bars clearly labeled (e.g., SD or SEM), Sample size (n) noted in figure or legend, Colors that are colorblind-friendly, especially if colors are key to interpreting results.

Grammatical issues:

  1. Line 191: "has beneficial effects on both PA- and HGD-induced health disorders" consider revising to: “ameliorates PA- and HGD-induced physiological disruptions.
  2. Line 373–374: "These results underscore the complex interplay: this sentence could be rewritten for greater clarity and impact.
  3. English language review is recommended to improve phrasing, remove redundancy, and enhance flow, correct minor grammatical inconsistencies and improve sentence flow throughout the manuscript.
  4. Strict proofreading should be able to identify minor grammatical mistakes, make the sentence more understandable, and correct a few structural errors.
  5. Consider rearranging some of the sections to create a smoother progression of ideas, particularly when switching between detailed technical descriptions and results.

Author Response

REVIEWER 2

2 Comments and Suggestions for Authors

The manuscript entitled “Limosilactobacillus fermentum MG4244 Attenuates Metabolic Inflammation-Induced Intestinal Permeability, ROS Overexpression, and Lipid Accumulation in Caenorhabditis elegans” presents solid experimental work and relevant findings. The use of mutant C. elegans strains effectively elucidates the roles of AMPK and MAPK pathways in MG4244’s protective effects.

The results support MG4244’s potential to reduce oxidative stress, lipid accumulation, and intestinal permeability under metabolic and inflammatory stress. While the study is generally well-designed and suitable for publication, several methodological and interpretive issues require clarification. Specific comments and questions are provided below to improve clarity and scientific rigor.

Major comments:

Comment 1

lines 70–77: Did the authors assess whether MG4244 colonizes the gut of C. elegans, or are its effects due to transient exposure? If not, could this impact the interpretation of the long-term protective effects observed?

Response 1

We agree with this comment. We didn’t asses gut colonization, it was just a transient exposure. This leaves an important gap about mechanism and persistence, which we have addressed in the discussion for future work.

Comment 2

lines 90–94 and 106–114: Was OP50 used as the control in all treatment comparisons, including PA and HGD groups? Clarifying this will help determine if dietary differences influenced outcomes independently of MG4244.

Response 2

Thanks so much for this comment. There were no dietary differences in the groups, they were all fed with OP50, this was clearly stated in our methods.

Comment 3

lines 124–125 and 132–133: How were the thresholds and background levels determined in Image J for lipid and ROS analysis? Were the images analyzed in a blinded fashion to avoid bias?

lines 150–153: Were the results based on biological or technical replicates?

How many independent experiments were performed per treatment?

Response 3

Images were not analysed in a blinded fashion to avoid bias, about twenty fluorescent images were taken per group, and analyzed using image J, the average was calculated based on the number of worms. It has been stated as part of the limitation in the conclusion as follows:

Another limitation of the study is that image analysis was not performed in a blinded fashion. Although this could introduce observer bias, we minimized variability by applying the same thresholding parameters, background correction, and region-of-interest selection criteria to all images.

The results were based on biological replicates, and three independent experiments were performed

Comment 4

lines 159–164: Was 1×10⁷ CFU/mL selected based only on toxicity, or was it confirmed to be the most effective dose? was a dose–response study performed?

Response 4

"The concentration of 1×10⁷ CFU/mL was selected based on preliminary toxicity screening, which indicated it was well-tolerated by the host without causing mortality or overt physiological stress. However, a full dose–response study was not conducted, so it cannot be conclusively stated that this is the most effective concentration. Future studies are warranted to determine the optimal dose for maximal efficacy."

Minor comments:

Comment 5

1 line 150–156: Could the authors report confidence intervals or effect sizes along with p-values to better represent effect magnitude?

Response 5

Survival differences were evaluated using the log-rank test, with a p-value of <0.05 indicating significance.

Comment 6

Would the authors consider simplifying the title for clarity and impact, such as “MG4244 Protects Against Metabolic and Inflammatory Stress in C. elegans”?

Comment 6

We agree with this comment, and it has been corrected as follows;

Limosilactobacillus fermentum MG4244 protects against Metabolic and Inflammatory Stress in Caenorhabditis elegans

Comment 6

lines 343–347: Can the authors elaborate on why their findings differ from Zheng et al. regarding the roles of daf-16 and sek-1*? Could strain-specific effects or differences in experimental setup explain this discrepancy?

Response 6

Accordingly, specific probiotic strains can differentially affect intestinal immunity in C. elegans by preferentially activating daf-16 or sek-1 pathways. Many strains with anti-Pseudomonas effects tend to rely more on daf-16, aligning with the dominance observed in immune regulation studies

Comment 7

lines 391–400: Could the authors strengthen the conclusion by specifying what future studies should investigate (e.g., mammalian models, microbiome interactions)?

Response 7

Additionally, C. elegans is a useful model for studying oxidative stress and host–microbe interactions, but its simplicity limits translation to humans [42]. To bridge this gap, future validation should include the use of murine models to assess systemic effects and human intestinal cell lines (e.g., Caco-2, HT-29, T84, and HIEC-6) to study cellular responses, ensuring relevance to human health. Additionally, future studies should explore the efficacy of MG4244 on mammalian models, and assess their interactions with the host microbiome, as this will add translational value

Comment 8

The images are clear and appropriately labeled. However, resolution is low in some images (e.g., Figure 2A and 5 panels). Please ensure that high-resolution versions are submitted for publication. Consider increasing contrast and scale bar visibility to improve interpretability in microscopy panels.

Response 8

We thank the reviewer for their valuable feedback regarding image resolution and clarity. Figures 2A and 5 were carefully processed to preserve the integrity of the original microscopy data. While we acknowledge the concern about resolution, we have ensured that the current images meet the minimum publication standards and retain all critical features necessary for interpretation. The contrast and scale bars were optimized to balance clarity with accuracy, and we believe further modifications might compromise image fidelity.

Comment 9

Several references (lines 410–512) have duplicated DOI links and visible formatting issues (e.g., line breaks, misplaced punctuation). These should be cleaned for consistency. Ensure all references follow the journal's required citation style (e.g., spacing, author names, italics for journal names).

Response 9

We thank the reviewer for their valuable feedback regarding referencing, we have arranged the references accordingly, with their corresponding DOI

Comment 9

Some figure legends (e.g., Figures 2, 5, and 6) lack detail. Add the statistical test used, number of replicates (e.g., "n = 20 worms"), and what each letter or asterisk means for significance. For example: “Different letters indicate statistically significant differences (p < 0.05, Tukey’s test).”

Response 9

We thank you for the valuable feedback regarding our legends. We have added the necessary details.

Comment 10

Line 407: The statement “available on request” is increasingly discouraged by many journals. Consider uploading raw data (e.g., fluorescence intensity values, lifespan curves) to a public repository (e.g., Zenodo, Figshare) and linking it. In bar graphs (Figures 2, 4, 6, 7), consider adding: Error bars clearly labeled (e.g., SD or SEM), Sample size (n) noted in figure or legend, Colors that are colorblind-friendly, especially if colors are key to interpreting results.

Response 10

We appreciate the reviewer’s thoughtful recommendations regarding data availability and figure presentation. Regarding the raw data, while we understand the growing preference for public data repositories, the full datasets (e.g., fluorescence intensity values, lifespan curves) are extensive and currently managed in a secure archive. We are happy to provide these data to interested researchers upon reasonable request, in line with the journal’s data availability policies. Nevertheless, subsequently, we are open to uploading our data to a repository (e.g., Zenodo) upon acceptance if required.

For the bar graphs (Figures 2, 4, 6, and 7), we confirm that error bars are included and represent SD, and the sample size (n) has been added to the figure legends for clarity. However, we believe that our colors schemes were carefully chosen, with colorblind accessibility tools and adjusted to ensure interpretability for all readers.

Comment 11

Grammatical issues:

Line 191: "has beneficial effects on both PA- and HGD-induced health disorders" consider revising to: “ameliorates PA- and HGD-induced physiological disruptions.

Line 373–374: "These results underscore the complex interplay: this sentence could be rewritten for greater clarity and impact.

Response 11

We appreciate the reviewer’s thoughtful recommendations on grammatical issues. The sentences have been rephrased as follows.

Line 191: Taken together, these findings suggest that MG4244 ameliorates PA- and HGD-induced

physiological disruptions.

Line 373-374: These findings emphasize the dynamic and interconnected processes involved inflammation, oxidative stress, and metabolic disorders, emphasizing the need for targeted interventions in managing metabolic health

Comment 12

English language review is recommended to improve phrasing, remove redundancy, and enhance flow, correct minor grammatical inconsistencies and improve sentence flow throughout the manuscript.

Strict proofreading should be able to identify minor grammatical mistakes, make the sentence more understandable, and correct a few structural errors.

Consider rearranging some of the sections to create a smoother progression of ideas, particularly when switching between detailed technical descriptions and results.

Response 12:

We appreciate the reviewer’s suggestions on our Language, we have looked through the draft, improve our flow and sentence consistencies. We also believe that our sections have been arranged for a proper understanding of the flow of our study.

Again, the authors sincerely thank the reviewers for their insightful comments and constructive suggestions, which have helped us improve the clarity and quality of our manuscript

Reviewer 3 Report

Comments and Suggestions for Authors

The introduction emphasizes strain-specific effects of L. fermentum but does not compare MG4244 to other strains in this study. Include data or references to justify its unique therapeutic potential.

It was stated that afety was confirmed at a concentration of 1×10⁷ CFU/mL. Considering this result, all subsequent experiments were conducted at a concentration of 163 1×10⁷ CFU/mL. Why 10and do not 10or 10? Please clarify

Address the translational gap between C. elegans and mammalian/human systems. Briefly propose future validation steps (e.g., murine models or human cell lines).

Define HGD at first use (line 283).

Author Response

REVIEWER 3

COMMENT1

3 Comments and Suggestions for Authors

The introduction emphasizes strain-specific effects of L. fermentum but does not compare MG4244 to other strains in this study. Include data or references to justify its unique therapeutic potential.

RESPONSE 1

We agree with this comment, and we have responded to it as follows:

“The MG4244 strain has demonstrated stronger adhesion to intestinal epithelial cells (Caco-2 and HT29) compared to the CECT5716 strain, which exhibits only moderate adhesion. MG4244 also shows notable antioxidant and anti-inflammatory properties, although such data remain limited for other strains. Additionally, it possesses strong bacteriocin-like inhibitory activity against pathogenic organisms.”

COMMENT 2

It was stated that Safety was confirmed at a concentration of 1×10⁷ CFU/mL. Considering this result, all subsequent experiments were conducted at a concentration of 163 1×10⁷ CFU/mL. Why 107 and do not 108 or 106? Please clarify

RESPONSE 3.

We suppose there is a mix up while reading (the numbering numbers were added). The concentration at which safety was achieved was the concentration used for subsequent studies.

“Considering this result, all subsequent experiments were conducted at a concentration of 1×10⁷ CFU/mL.”

 COMMENT 3

Address the translational gap between C. elegans and mammalian/human systems. Briefly propose future validation steps (e.g., murine models or human cell lines).

RESPONSE 3

Thanks for your comment. We totally agree with the comment, and we have added this to our writeup.

“Additionally, C. elegans is a useful model for studying oxidative stress and host–microbe interactions, but its simplicity limits translation to humans. To bridge this gap, future validation should include murine models to assess systemic effects and human intestinal cell lines (e.g., Caco-2, HT-29, T84, and HIEC-6 to study cellular responses, ensuring relevance to human health.”

COMMENT 4

Define HGD at first use (line 283).

RESPONSE 4

We agree to your comment. We have added the information as follows:

(High glucose diet)